# Pregnancy and neonatal outcomes in COVID-19: study protocol for a global registry of women with suspected or confirmed SARS-CoV-2 infection in pregnancy and their neonates, understanding natural history to guide treatment and prevention

Jayanta Banerjee [1,2] Edward Mullins [2,3] Julia Townson,[4] Rebecca Playle,[4] Caroline Shaw,[2,3] Nigel Kirby,[4] Kim Munnery,[4] Tom Bourne,[2] TG Teoh,[2,5] Mandish Dhanjal,[5] Liona Poon,[6] Alison Wright,[7] Christoph Lees [2,3,5]

JB and EM are joint first authors.

For numbered affiliations see end of article.

**Correspondence to**
Professor Christoph Lees;
c.lees@imperial.ac.uk

## ABSTRACT

**Introduction** Previous novel COVID-19 pandemics, SARS and middle east respiratory syndrome observed an association of infection in pregnancy with preterm delivery, stillbirth and increased maternal mortality. COVID-19, caused by SARS-CoV-2 infection, is the largest pandemic in living memory.

Rapid accrual of robust case data on women in pregnancy and their babies affected by suspected COVID-19 or confirmed SARS-CoV-2 infection will inform clinical management and preventative strategies in the current pandemic and future outbreaks.

**Methods and analysis** The pregnancy and neonatal outcomes in COVID-19 (PAN-COVID) registry are an observational study collecting focused data on outcomes of pregnant mothers who have had suspected COVID-19 in pregnancy or confirmed SARS-CoV-2 infection and their neonates via a web-portal. Among the women recruited to the PAN-COVID registry, the study will evaluate the incidence of: (1) miscarriage and pregnancy loss, (2) fetal growth restriction and stillbirth, (3) preterm delivery, (4) vertical transmission (suspected or confirmed) and early onset neonatal SARS-CoV-2 infection.

Data will be centre based and collected on individual women and their babies. Verbal consent will be obtained, to reduce face-to-face contact in the pandemic while allowing identifiable data collection for linkage. Statistical analysis of the data will be carried out on a pseudonymised data set by the study statistician. Regular reports will be distributed to collaborators on the study research questions.

**Ethics and dissemination** This study has received research ethics approval in the UK. For international centres, evidence of appropriate local approval will be required to participate, prior to entry of data to the database. The reports will be published regularly. The outputs of the study will be regularly disseminated to participants and collaborators on the study website

## Strengths and limitations of this study

► The pregnancy and neonatal outcomes in COVID-19 (PAN-COVID) registry aims to collect an international data set, which will enable us to answer a focused set of questions related to pregnancy, maternal and neonatal outcomes of SARS-CoV-2 infection in pregnancy.

► There is a high likelihood of an adequate sample size to allow us to measure all the stated focused outcomes of SARS-CoV-2 infection in pregnancy.

► The PAN-COVID protocol facilitates data sharing and collaboration with multiple global partners.

► This registry establishes a collaborative framework, which can be used to respond effectively to guide care for women in pregnancy and their neonates in future pandemics.

► Registry studies are generally unlikely to fully capture all cases and therefore carry risk of underestimating infection rates and overestimating infection complication and fatality rates.

(https://pan-covid.org) and social media channels as well as dissemination to scientific meetings and journals.

**Study registration number** ISRCTN68026880.

## INTRODUCTION

The majority of all new, emerging or re-emerging viruses affecting the human population in the recent years are zoonoses, including HIV/AIDS, SARS, the H5N1 strain of avian influenza, the 2009 pandemic H1N1 influenza virus, the middle east respiratory syndrome (MERS), the Zika and Nipa viruses[1 2] and most recently COVID-19

caused by SARS-CoV-2 virus. The spread of these infections in an immune-naïve human population, as shown by the recent COVID-19 pandemic poses serious strain on public health. To tackle such large pandemics, international collaborative and multidisciplinary approaches are necessary.

These should aim to identify those at highest risk of exposure to emerging pathogens, to characterise the culture and practices that modify their risk (eg, the presence of live animal markets and the use of face masks in South-East Asia) and to develop preventative interventions accordingly.[3 4] Pregnant women infected with SARS or MERS were at increased risk of mortality and morbidities such as stillbirth, preterm birth and fetal growth restriction (FGR).[5 6] Case series for pregnant women infected during outbreaks of SARS[7] or MERS[8] report a variety of different outcomes, with variable recording of diagnostic testing, maternal, fetal and neonatal outcomes and the presence or absence of vertical transmission. Clinical outcomes appear worse for pregnant compared with non-pregnant women infected with SARS and H1N1 influenza.[7 9]

As of the 12 May 2020, after disambiguation and removal of duplicate reporting, the rolling review of all cases worldwide conducted by Thornton et al[10] contains 92 publications including details of 806 pregnant women and 674 fetuses. At the time of writing this protocol, 457 of the included women had delivered and were alive and 21 women had died. These women had delivered 428 babies, of which 94 were born preterm, 17 were infected with COVID-19, and there were 15 neonatal deaths. The largest study to date has been from the UK (Knight et al) who reported 427 cases (247 completed pregnancies): there were five (1%) maternal deaths, three (1%) stillbirths and two (1%) neonatal deaths.[11] There are currently limited data on the effect of COVID-19 on second trimester miscarriage[12] and no available data on first trimester miscarriage or growth restriction. The risk of reported probable vertical transmission of SARS-CoV-2 infection is estimated at 2.5% in women admitted to hospital with COVID-19; with case reports describing increased SARS-CoV-2 IgM levels in neonates, and pneumonia and lymphopaenia in neonates with negative COVID-19 reverse transcription-PCR testing, principally from nasopharyngeal swabs.[13 14]

There are major knowledge gaps on the effect of COVID-19 on various stages of pregnancy, and its effect on fetuses in terms of growth restriction, prematurity and short and long term morbidities.[15] There is an urgent need to collect case data rapidly, to pool global data on the natural history of women affected by suspected COVID-19 or confirmed SARS-CoV-2 in pregnancy to inform treatment and implement preventative strategies in the current and future outbreaks. Published case series are almost always out of date when published and cases may overlap. A centre-based registry, gathering case data prospectively on the effect of SARS-CoV-2 infection from healthcare systems around the world offers a method to accrue clinical outcomes on key research questions from a variety of populations and healthcare systems without these limitations.

The pregnancy and neonatal outcomes of COVID-19 (PAN-COVID) registry will focus on miscarriage, FGR, stillbirth, preterm delivery and vertical transmission (suspected or confirmed) and early onset neonatal SARS-CoV-2 infection and will include fields on ultrasound diagnosis and neonatal care not included in other more general studies. This study will identify cases that should also be reported to International Network of Obstetric Survey Systems (INOSS) population-based surveillance systems of confirmed COVID-19 cases admitted to hospital through existing reporting mechanisms. The UK's obstetric surveillance system (UKOSS) is a population surveillance study, reporting women in pregnancy hospitalised with COVID-19 or SARS-CoV-2 infection. The PAN-COVID registry is distinct in that it will collect, via a web portal, data on SARS-CoV-2 infections in pregnancy, whether the diagnosis is presumptive based on symptoms or following a positive test and whether the patient is admitted to hospital or not. Given that the SARS-CoV-2 virus is likely to affect a high proportion of the global population and adverse outcomes may influence policy and practice in a short time frame, 'real-time' high-level reporting on a regular basis will be valuable for UK and international clinicians and policymakers.

PAN-COVID registry will work collaboratively with approved studies and registries to ensure that the global impact of the pandemic is captured as fully as possible, with data sharing and linkage. PAN-COVID has established links with the UK National Neonatal Research Database (NNRD) (REC Reference: 16/LO/1093) who will provide data on neonatal outcomes, and the British Paediatric Surveillance Unit with whom there is in principle an agreement for data sharing. In the UK, the PAN-COVID registry will signpost to UKOSS where the case definition requires a more detailed data collection. We are in discussions with international studies including the American Academy of Pediatrics (AAP) Section of Neonatal Perinatal Medicine (SONPM) to agree on a common data set to enable future merging of information.

## METHODS AND ANALYSIS

This is an observational global pregnancy and neonatal register (PAN-COVID registry) collecting outcome data from women in pregnancy who have confirmed SARS-CoV-2 infection or signs and symptoms of COVID-19 during their pregnancy. The study is sponsored by Imperial College London and funded by UK Research Institute and National Institute of Health and Research (NIHR). The study received Urgent Public Health (UPH) prioritisation in the UK in April 2020.

The main study objectives are, first, to establish a UK and international disease registry for women with suspected COVID-19 or confirmed SARS-CoV-2 infection in pregnancy; in the UK, this will be linked to neonatal data

(from NNRD). Second, PAN-COVID investigators aim to publish regular reports focused on our principle research question: in women recruited to the PAN-COVID registry with suspected or confirmed SARS-CoV-2 infection, what is the incidence of (a) miscarriage, (b) small for gestational age (SGA)/FGR and stillbirth, (c) preterm birth and (d) suspected or confirmed vertical transmission to the neonate and perinatal infection.

## Data entry

Demographics—date of birth, National Health Service (NHS) number (or international equivalent), smoking status, last menstrual period, expected date of delivery, body mass index, history of previous pregnancies/miscarriages/loss/small for gestational age/delivery (preterm/term), comorbidities, medications, ethnicity, number of fetuses, structural abnormality on ultrasound.

COVID-19 signs and symptoms—fever, new persistent cough, anosmia, myalgia, diarrhoea, shortness of breath, fatigue, abdominal pain, chest pain, hoarseness of voice, loss of appetite, delirium.

SARS-CoV-2 infection—SARs-CoV 2 test performed, date of test(s), results of test(s) (if available)

COVID-19 treatment—requirement for (including duration of) inpatient, level 2 or 3 critical care, supplementary oxygen, non-invasive ventilation, endotracheal intubation, extracorporeal membrane oxygenation, antiviral/other specific anti-COVID-19 therapy, iatrogenic delivery or termination due to maternal compromised from COVID-19.

Maternal outcomes—non-invasive ventilation, intubation and ventilation and maternal death.

Comorbidities—hypertension, respiratory disease, cardiovascular disease, renal disease, autoimmune disease, medications including aspirin, progesterone, others.

Delivery details—iatrogenic/spontaneous, date of delivery/miscarriage, mode of delivery, live birth/stillbirth/miscarriage, birth weight, sex.

Postnatal outcomes—breastfeeding, baby cared for separately from mother until discharge.

Neonatal outcomes—date of birth, birth weight, gestation at birth, Apgar at 5 min, neonatal death, COVID-19 transmission (neonate with positive swab/serology), congenital abnormality/deformation, date of discharge from hospital, readmission up to 28 days of life.

## Study population

Study centres will be asked to identify eligible women of childbearing age (18–50 years) at the time of an early pregnancy attendance or at delivery who have had suspected COVID-19 or confirmed SARS-CoV-2 infection in pregnancy. Case data will be entered into the online registry. The study will recruit women affected by suspected COVID-19 or confirmed SARS-CoV-2 infection from January 2020 and March 2021. Participants will be able to withdraw from the register at any time without affecting their medical care. Participants should

inform any member of the study team of their wish to withdraw consent. When a participant withdraws consent, they should be asked whether they are willing to allow ongoing data collection, such as maternal and fetal outcomes, from their medical records. If the participant withdraws consent and does not wish to have any further data collected, then this will be recorded on the database and no further data entry will be possible.

## Outcomes

Assessment of outcomes will require follow-up by individual healthcare professionals, by accessing medical records routinely available to them as part of the clinical care team. When a pregnant woman with SARs-CoV-2 infection or suspected COVID-19 is registered on the database, they will automatically be assigned a unique participant identification number. Date of birth and NHS number, or equivalent in non-UK centres, will be stored on the database. The limit for data collection will be 28 days after the delivery or pregnancy loss of the last woman registered. Linkage will allow data up to 2 years of age to be collected for preterm babies in the UK with the NNRD database.

## Data collection pathway

The Centre for Trials Research (CTR), Cardiff University, has designed a registry and web portal for data entry for healthcare professionals. The CTR hosts and maintains the study web page (https://pan-covid.org), which allows maternity healthcare providers to register interest in submitting data to an online database. The data manager will manage the registration of healthcare professionals, keeping personal identifiable information on a separate, password protected, spreadsheet with restricted access. The study co-ordinator will manage registration of healthcare professionals and work with the data manager to provide login permission to those who wish to contribute to the register. Login permission will only be granted to healthcare professionals who have confirmed their email address, which must be from a bona fide healthcare or educational organisation and local agreement and ethical approval is in place.

Where a web portal cannot be accessed by the investigators, a bespoke spreadsheet has been designed to allow automatic uploading of data to the main registry database.

## Analysis

Statistical analysis of the data will be carried out a pseudonymised data set containing NHS number (or equivalent international data field). Numerical data for cases reported and outcomes (miscarriage, SGA/FGR, stillbirth, preterm delivery and vertical transmission) will be reported in anonymised, aggregate regular reports, published on the study website.

## Sample size

Prespecified sample size estimation was not carried out given the aim of this observational study was to collate

all consecutive eligible case outcomes in participating centres between data collection start and 18 months from start of data collection. It may be useful to provide an approximate guide as to the width of the 95% CI that can be achieved for proportions using historical data from the UK. These historical data may not hold during the current pandemic and may alter due to multiple reasons such as access to healthcare or behaviour changes. Contemporaneous data collected using other current national registries will provide true background rates for comparative purposes at the end of the study period. Generalising these data internationally may not be possible unless similar contemporaneous data exist for comparison.

The estimated representative pre-COVID pandemic incidences of miscarriage, Small for Gestational Age (SGA)/FGR and stillbirth in the UK were 30%, 10%, 0.2%, respectively. The expected outcome proportions during the COVID-19 pandemic are 40%, 15% and 0.4%, respectively. A sample size of 500 would allow the estimation of the width of 95% CIs for the proportion of miscarriage as 40%±4.2%, for SGA/FGR 15%±2.9% and for stillbirth 0.4%±0.3%.

These figures are meant only as a guide and reference proportions may change over this period. Since attaining sufficient precision for stillbirth requires by far the largest sample size, then study recruitment targets could be based on this rarest of outcomes. This would then provide sufficient precision to assess other more common outcomes. These figures are based on data from the UK only. Reference proportions vary by country but are of a similar order to those in the UK.

Appropriate quantitative analyses will be conducted by a study statistician. Dates collected from sites for expected data of delivery or last menstrual period, week of pregnancy at delivery and date of any COVID-19 tests in the participant or neonate will be cross-checked and validated for the final report. Regular interim reports will use contemporaneous data as it is collated.

Derived variables to be reported are:
► Infection delivery interval (<2 or≥2 weeks after onset of suspected or confirmed COVID-19).
► Week of pregnancy when COVID-19 diagnosis was made.
► Number of days from delivery to SARS-COV-2 test
► Birth weight percentiles adjusted for gender and gestational age.

Fortnightly bulletins will consist of counts of participants by country, proportions for maternal COVID-19 diagnosis status (suspected, positive test, negative test), week of pregnancy for COVID-19 diagnosis (<23 weeks, 23–36 weeks, ≥37 weeks), weeks of pregnancy at delivery (<23 weeks, 23–36 weeks, ≥37 weeks), maternal and neonatal mortality, outcome of delivery (live birth, miscarriage, intrauterine death, stillbirth (>22+6 weeks gestation)), FGR split by delivery interval (<2 vs ≥2 weeks), COVID-19 diagnosis status for babies split by number of days from delivery to test (0, 1–7, 8–28) and birth weight percentiles. Mean (SD) and median birth weights will

also be reported. Further analysis will take place at one interim time point and at final data collection. For the interim analysis, a subset of validated outcomes will be reported consisting of total cohort proportions of gestation at delivery, birth weight Z-scores, perinatal mortality (stillbirth, early neonatal death), neonatal COVID-19 or SARS-COV-2 swab positive, neonatal morbidity and maternal death. These data will be split by suspected and confirmed COVID-19 diagnosis. Additional graphical exploratory analysis of the data will assess any possible association between the infection delivery interval and the timing of delivery as well as birth weight. The final analysis will report all demographic baseline data for the whole cohort and all outcomes as well as these graphical trends. Proportional data for the study outcomes will be presented with 95% CIs, by country, where numbers allow. Data and all appropriate documentation will be stored for a minimum of 10 years after the completion of the study, including the follow-up period.

## ETHICS AND DISSEMINATION

The Study Coordination Centre has obtained approval from the Haydock Research Ethics Committee (REC) and Health Regulator Authority, REC reference: 20/NW/0212 and had been considered to be an Urgent Public Health study by the UK Clinical Research Network.

The study must also receive confirmation of capacity and capability from each participating NHS Trusts in the UK before accepting participants into the study or any research activity is carried out. The study will be conducted in accordance with the recommendations for physicians involved in research on human subjects adopted by the 18th World Medical Assembly, Helsinki 1964 and later revisions. The international participating centres will be required to obtain appropriate local approval from their centres. All those taking consents will be qualified and trained to take consent in a medical context. To reduce face-to-face consultations during the pandemic verbal consent will be obtained after provision of the participant information sheet: Patient Information Sheet (PIS) (online supplemental appendix 1), which will have details of how to contact the PAN-COVID study team, if they would like further information or to withdraw from the study and this will be recorded in the patients notes and on the study database. Where women have lost capacity to provide consent and have confirmed COVID-19, women will be assumed to provide consent. If they regained capacity, the study will be explained to them and they will be provided with the PIS, which will have details of how to contact the PAN-COVID study team, if they would like further information or to withdraw from the study. If they choose to withdraw, data pertaining to their care would be removed from the registry. Where feasible, the woman's partner or a family member will be provided with the PIS and asked if they feel able to advise on the presumed wishes and feelings of participants unable to consent for themselves and on

their inclusion in the research. Their response will be recorded on the database and in the participants' clinical notes. These models will give a reasonable opportunity for women to provide their data to the study, while being mindful not to over burden healthcare resources. Research has shown that within a pandemic, potential participants and their families are broadly agreeable to alternative methods of gaining consent, therefore we decided on this verbal model of consent, as a pragmatic solution to ensure women are informed of how their data will be used and alleviate additional burden to healthcare resources.[16] The chief investigators (CL and EM) will preserve the confidentiality of participants taking part in the study and is registered under the General Data Protection Regulation (GDPR) 2018. In terms of indemnity, Imperial College London holds negligent harm and non-negligent harm insurance policies that apply to this study. Imperial College London will act as the Sponsor for this study. Delegated responsibilities will be assigned to the NHS trusts taking part in this study. The study may be subjected to inspection and audit by Imperial College London under their remit as sponsor and other regulatory bodies to ensure adherence to Good Clinical Practice (GCP) and the UK Policy Framework for Health and Social Care Research.

### Data security

Cardiff University has strict policies and processes in place to ensure adherence to the GDPR[2] The University is registered on the Information Commissioner's Office Data Protection register, number Z6549747. All data collected on the Registry will be held on Cardiff University servers, which have a high level of security. CTR database developers have run a security check on the trials.cardiff.ac.uk server and it has passed to an acceptable standard to EU and non-EU centres.

The person allocating centre IDs and logins will not have access to the database fields containing personal information. For healthcare professionals to enter follow-up data, they will need to maintain a secure spreadsheet (password protected) with participants' identifiable information logged against their unique participant identification number.

To reduce any potential risk of identification of study participants, access will be restricted to the section of the database containing identifiable information (NHS number, date of birth, ethnicity) to those responsible for maintaining the database (two individuals). This will mean that the CTR members of the research team (two different individuals) responsible for allocating the unique centre and participant IDs will not be able to access any identifiable data. In addition, all other CTR members of the research team will be blinded to which centre ID relates to which centre. For statistical analyses, a data set will be prepared by the database developers which will have all NHS numbers, and the international equivalent, removed, reducing possible identification of participants further.

### Patient and public involvement

Patients and public were not involved in the initial research idea and study design, due to the rapid response required at the start of the COVID-19 pandemic and lockdown. The NIHR funded this research and involves patients and public in setting research priorities and in reviewing funding applications. Patients and the public will be notified of the results through our established links within the UK including the Royal College of Obstetrics and Gynaecology (RCOG), International Society of Ultrasound in Obstetrics and Gynaecology (ISUOG), International Federation of Gynaecology and Obstetrics (FIGO) and related partner charities and with our US partner the AAP. We expect that the major publications from this study will be widely cited in the scientific and lay press and this will prove to be the most effective route of dissemination.

### Dissemination

Regular reports of case data will be made available on the study website. Although members of the public were not involved in the study design as it is broadly a registry, the maternity and neonatal focus groups will be involved in the dissemination of the outputs. The outputs of the PAN-COVID registry will be presented in national and international conferences, lay and digital social media and published in peer-reviewed scientific journals which will be made available as open access, as per the NIHR's publication policy.

**Author affiliations**
[1]Department of Neonatology, Imperial College Healthcare NHS Trust, London, UK
[2]Metabolism, Digestion and Reproduction, Faculty of Medicine, Imperial College London, London, UK
[3]Centre for Fetal Care, Queen Charlotte's and Chelsea Hospital, Imperial College Healthcare NHS Trust, London, UK
[4]Centre for Trials Research, Cardiff University, Cardiff, UK
[5]Women, Children and Clinical Support, Imperial College Healthcare NHS Trust, London, UK
[6]Obstetrics and Gynaecology, The Chinese University of Hong Kong, Hong Kong, Hong Kong
[7]Obstetrics and Gynaecology, Royal Free London NHS Foundation Trust, London, UK

**Contributors** CL and EM conceived of the study and prepared the study design. JB, TB, CS and JT helped in preparing the study design. JB prepared the first draft. RP, NK and KM helped in the statistical design of the registry, preparation of the online platform for recruitment and writing the method section of the manuscript. TT, MD, LP and AW helped in development of the study design and preparation of the manuscript. All authors contributed to the manuscript and agreed the final draft. All authors agreed to be accountable for all aspects of the work in ensuring that questions related to the accuracy or integrity of any part of the work are appropriately investigated and resolved.

**Funding** The PAN-COVID registry is funded by the United Kingdom Research Institute (UKRI) and National Institute of Health and Research (NIHR) through COVID-19 Rapid Response Call 2, grant reference MC_PC 19066.

**Competing interests** None declared.

**Patient consent for publication** Not required.

**Provenance and peer review** Not commissioned; externally peer reviewed.

responsibility arising from any reliance placed on the content. Where the content includes any translated material, BMJ does not warrant the accuracy and reliability of the translations (including but not limited to local regulations, clinical guidelines, terminology, drug names and drug dosages), and is not responsible for any error and/or omissions arising from translation and adaptation or otherwise.

**ORCID iDs**
Jayanta Banerjee http://orcid.org/0000-0002-7765-8439
Edward Mullins http://orcid.org/0000-0003-1886-6358
Christoph Lees http://orcid.org/0000-0002-2104-5561

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
