## [Reviewer comments · BMJ Open]

ARTICLE DETAILS

TITLE (PROVISIONAL)	Pregnancy and Neonatal Outcomes in COVID-19: Study protocol for a global registry of women with suspected or confirmed SARS-CoV-2 infection in pregnancy and their neonates, understanding natural history to guide treatment and prevention
AUTHORS	Banerjee, Jayanta; Mullins, E; Townson, Julia; Playle, Rebecca; Shaw, Caroline; Kirby, Nigel; Munnery, Kim; Bourne, Tom; Teoh, Tg; Dhanjal, Mandish; Poon, Liona; Wright, Alison; Lees, Christoph

VERSION 1 – REVIEW

REVIEWER	Esau Joao Hospital Federal dos Servidores do Estado, Brazil
REVIEW RETURNED	01-Jul-2020

GENERAL COMMENTS	Congratulations on this important initiative that attempts to standardize data on pregnant women with COVID-19. I am certain that the outcomes of this study will be very important for resolving unanswered questions about COVID infection during pregnancy. Another strength of this initiative is promoting the participation of international sites all over the world. I encourage the authors to involve sites in countries where the pandemic is at its peak such as the United States, Latin American countries, and Africa. One of my concerns is that as the study plans to recruit from January 2020 to March 2021, there will be data retrospective and prospective data. In light of this, the analysis will need to be planned carefully to reconcile data from different sites so as to standardize all of the variables. With respect to sample size, the prevalence calculation assumes that the participants are independent. As there will likely be correlation among women (e.g. by country), the sample size would need to be increased to account for this design effect. As a suggestion to the authors, you might want to consider whether it would be possible to implement a case-control design, as this would make the study's findings become more robust.
--

REVIEWER	Eva Bermejo-Sánchez, PhD Institute of Rare Diseases Research (IIER), Instituto de Salud Carlos III (ISCIII). Spain
REVIEW RETURNED	12-Aug-2020

GENERAL COMMENTS	This study protocol for a global registry of registry of women with suspected or confirmed SARS-CoV-2 infection in pregnancy and their neonates, refers to an observational study for which the research questions and outcomes are clearly defined, the study
--

	design seems appropriate to answer those research questions, and the methods are described sufficiently, although the protocol would benefit from a more detailed description of the statistical analyses to be carried out. For such analyses, it could be planned to perform separate and comparative calculations for suspected COVID-19 and for confirmed SARS-CoV-2 infection. This is an extremely important and needed study, in which a huge sample size could be achieved on an international basis, aimed to clarify whether suspected COVID-19 or confirmed SARS-CoV-2 infection are related to miscarriage and pregnancy loss, fetal growth restriction, stillbirth, pre-term delivery or vertical transmission. Data from this study can be fully insightful to these respects, what will help in the decision making process and to inform pregnant women more accurately.
--	--

VERSION 1 – AUTHOR RESPONSE

Response to Reviewer(s)' Comments to Author:

Response to comments of Reviewer: 1

Congratulations on this important initiative that attempts to standardize data on pregnant women with COVID-19. I am certain that the outcomes of this study will be very important for resolving unanswered questions about COVID infection during pregnancy. Another strength of this initiative is promoting the participation of international sites all over the world.

Response: Thank you we agree that this is a worthwhile project and is accordingly supported by the UK Urgent Public Health (UPH) programme.

I encourage the authors to involve sites in countries where the pandemic is at its peak such as the United States, Latin American countries, and Africa.

Response: We are recruiting from a large global population and have forged links with US, Brazil, Peru, Argentina, Canada, European, African and Australasian countries.

One of my concerns is that as the study plans to recruit from January 2020 to March 2021, there will be data retrospective and prospective data. In light of this, the analysis will need to be planned carefully to reconcile data from different sites so as to standardize all of the variables.

Response: The statistical analysis for this registry is planned with this in mind, though it will not be possible to standardize for all variables given the wide differences in diagnosis of the condition and ascertainment of outcomes. We will use descriptive statistics to compare major outcomes between different geographies. If numbers allow, we will include an exploration of time trends within the data set for retrospective vs prospective data collection. Please see page 11-12, paragraphs 3, 4 and 5 in the main tracked manuscript.

With respect to sample size, the prevalence calculation assumes that the participants are independent. As there will likely be correlation among women (e.g. by country), the sample size would need to be increased to account for this design effect.

Response: There will be correlation at many levels within this dataset but since we are not attempting any comparative analyses, as would be the case for an experimental design, sample sizes were provided as a guide only. We aim to maximise the size of our observational cohort sample within each country and will report outcome incidence for the whole cohort and by country where sample sizes are adequate. We have amended the sample size section to avoid confusion. Please see page 10, paragraph 4, line 4 in the main tracked manuscript.

As a suggestion to the authors, you might want to consider whether it would be possible to implement a case-control design, as this would make the study's findings become more robust.

Response: The registry is designed to evaluate the incidence of: 1. Miscarriage and pregnancy loss, 2. FGR and stillbirth, 3. Pre-term delivery and 4. Vertical transmission (suspected or confirmed) and early-onset neonatal SARS-CoV-2 infection.

The registry received funding from the NIHR and UKRI and approval from UK UPH based on the design that we set out in the protocol and primary analyses are pre-planned. A case control design is an interesting and excellent idea, though a prospective study of this type would need very significant resource to allow it to work; we are unable to do this within our current design or funding envelope.

Response to comments of Reviewer: 2

This study protocol for a global registry of registry of women with suspected or confirmed SARS-CoV-2 infection in pregnancy and their neonates, refers to an observational study for which the research questions and outcomes are clearly defined, the study design seems appropriate to answer those research questions, and the methods are described sufficiently, although the protocol would benefit from a more detailed description of the statistical analyses to be carried out. For such analyses, it could be planned to perform separate and comparative calculations for suspected COVID-19 and for confirmed SARS-CoV-2 infection.

Response: The statistical analysis for this registry is pre-planned and public health experts and statisticians will analyse the data following recruitment of women. The statistical methods section of the manuscript has been updated with the derivation of variables to be reported. Please see page 10, paragraph 4, line 4 in the main tracked manuscript.

This is an extremely important and needed study, in which a huge sample size could be achieved on an international basis, aimed to clarify whether suspected COVID-19 or confirmed SARS-CoV-2 infection are related to miscarriage and pregnancy loss, fetal growth restriction, stillbirth, pre-term delivery or vertical transmission. Data from this study can be fully insightful to these respects, what will help in the decision-making process and to inform pregnant women more accurately.

Response: Thank you for your comments. We strongly believe that this is a very important study to answer the research questions mentioned above and the rapid accrual of robust case data on women in pregnancy and their babies affected by suspected COVID-19 or confirmed SARS-CoV-2 infection will inform clinical management and preventative strategies in the current pandemic and future outbreaks